SCIENTIFIC CORRESPONDENCE

# Comment on 'The clinical pharmacology of tafenoquine in the radical cure of *Plasmodium vivax* malaria: An individual patient data meta-analysis'

Raman Sharma[1], Chao Chen[2], Lionel Tan[2], Katie Rolfe[1], Ioana-Gabriela Fiţa[2], Siôn Jones[2], Anup Pingle[3], Rachel A Gibson[1]*, Navin Goyal[4†], Hema Sharma[2‡], Panayota Bird[2]

[1]GSK, Stevenage, United Kingdom; [2]GSK, Brentford, United Kingdom; [3]GSK, Mumbai, India; [4]GSK, Upper Providence, United States

**Abstract** A single 300 mg dose of tafenoquine, in combination with chloroquine, is currently approved in several countries for the radical cure (prevention of relapse) of *Plasmodium vivax* malaria in patients aged ≥16 years. Recently, however, Watson et al. suggested that the approved dose of tafenoquine is insufficient for radical cure, and that a higher 450 mg dose could reduce P. vivax recurrences substantially (Watson et al., 2022). In this response, we challenge Watson et al.'s assertion based on empirical evidence from dose-ranging and pivotal studies (published) as well as real-world evidence from post-approval studies (ongoing, therefore currently unpublished). We assert that, collectively, these data confirm that the benefit–risk profile of a single 300 mg dose of tafenoquine, co-administered with chloroquine, for the radical cure of *P. vivax* malaria in patients who are not G6PD-deficient, continues to be favourable where chloroquine is indicated for *P. vivax* malaria. If real-world evidence of sub-optimal efficacy in certain regions is observed or dose-optimisation with other blood-stage therapies is required, then well-designed clinical studies assessing safety and efficacy will be required before higher doses are approved for clinical use.

**\*For correspondence:**
Rachel.a.gibson@gsk.com

**Present address:** †Johnson & Johnson, New York, United States; ‡AstraZeneca, Cambridge, United Kingdom

## Introduction

The *Plasmodium vivax* malarial parasite has a major economic and public health impact, especially in regions such as East Africa, Latin America and South and East Asia (*GSK, 2018a*; *Hounkpatin et al., 2019*). When present in blood, *P. vivax* can cause acute malaria with episodes of chills, fever, muscle pains and vomiting. The parasite also has a dormant liver hypnozoite stage, which can reactivate after weeks, months or years, leading to relapses and, potentially, to severe anaemia, permanent brain damage and death (*GSK, 2018a*; *Hounkpatin et al., 2019*). For effective treatment, eradication of both the blood and liver stages of *P. vivax* is required (radical cure) (*Hounkpatin et al., 2019*).

Since 2018, regulators from the United States initially, and subsequently from Australia, Brazil, Colombia, Thailand, Peru and The Philippines, have approved tafenoquine (as a single oral dose of 300 mg in combination with standard doses of chloroquine) for the radical cure (prevention of relapse) of *P. vivax* malaria in patients aged ≥16 years (*GSK, 2018a*; *GSK, 2018b*; *Medicines for Malaria Venture, 2021* ). A paediatric formulation that allows weight-band-based dosing of children (aged ≥2 years) and adolescents is also approved in Australia (since 2022) (*Medicines for Malaria Venture, 2022*). Like primaquine, tafenoquine is an 8-aminoquinoline derivative effective against hypnozoites and all other stages of the *P. vivax* lifecycle; however, although the World Health Organization (WHO) recommends a 7- or 14 day treatment course for primaquine, tafenoquine is the first

single-dose treatment for the radical cure of *P. vivax* malaria and therefore has patient adherence and convenience advantages (*GSK, 2018a*; *GSK, 2018b*; *World Health Organization, 2023*). Nonetheless, as an 8-aminoquinoline, the safety profile of tafenoquine is similar to that of primaquine, and both agents can cause oxidant haemolysis in people with glucose-6-phosphate dehydrogenase (G6PD) deficiency (*Milligan et al., 2019*; *Llanos-Cuentas et al., 2019*). Acute haemolysis is usually short-lived and does not need specific treatment; however, in rare cases, severe haemolysis may lead to life-threatening anaemia (requiring red blood cell transfusions) or haemoglobinuric renal failure (*Baird, 2019*). In malaria-endemic regions it has been estimated that 8% of the population are G6PD deficient, although significant variation is reported across regions, with the highest country-specific prevalence estimated in Africa and Western Pacific countries (*Howes et al., 2012*; *P. vivax information hub, 2012*). G6PD deficiency is an X-linked disorder; males are either G6PD deficient or have normal G6PD activity, whereas females exhibit a wide range of G6PD deficiency (*Hounkpatin et al., 2019*). Females may be symptomatic if they are homozygous, or if they are heterozygous and inactivation of their normal X chromosome (lyonisation) is skewed towards a deficient phenotype (*Hounkpatin et al., 2019*; *Domingo et al., 2019*). Caution is needed because inter-individual variability in the pattern of lyonisation may cause heterozygous females with levels of enzyme activity between 30% and 70% of normal to test as normal for G6PD deficiency using qualitative, phenotypic, rapid diagnostic screening tests (*Chu et al., 2017*; *Baird et al., 2015*). To reduce the risk of haemolysis, the G6PD status of all potential tafenoquine patients must be determined with a quantitative test capable of accurately differentiating deficient, intermediate and normal G6PD activity levels, and tafenoquine should be withheld from patients with G6PD enzyme levels below 70% of normal (*GSK, 2018b*).

Importantly, appropriate clinical practice for the use of 8-aminoquinolines in *P. vivax* malaria has always been precariously balanced between providing adequate activity against hypnozoites and the real risk of haemolytic harm to patients with G6PD deficiency (*Shanks, 2022*). The cautious benefit–risk balance involved with the single 300 mg dose of tafenoquine has been questioned in a recently published paper in which Watson et al. hypothesise that the current recommended dose of tafenoquine 300 mg co-administered with chloroquine is insufficient and that a 450 mg dose of tafenoquine in this same setting would reduce the risk of relapse (*Watson et al., 2022*). That dose is 50% greater than the 300 mg dose approved by the US Food and Drug Administration (FDA), Australian Therapeutic Goods Administration (TGA) and other international regulatory authorities (*GSK, 2018a*; *GSK, 2018b*; *Medicines for Malaria Venture, 2021*; *Medicines for Malaria Venture, 2022*). Herein, we discuss concerns regarding the conclusions of Watson et al.

- The benefit–risk profile of tafenoquine 450 mg is not appropriately considered. For example, there is minimal discussion of tafenoquine safety data and key findings from a phase 1 study in healthy female volunteers heterozygous for the G6PD Mahidol variant. This important study demonstrated not only that the haemolytic potential of tafenoquine was dose-dependent but also that a single 300 mg dose of tafenoquine had the same potential to cause haemolytic harm as the WHO-recommended dose of primaquine for uncomplicated *P. vivax* malaria (15 mg/day for 14 days) (*Rueangweerayut et al., 2017*; *World Health Organization, 2015*).
- We acknowledge that data from the phase 2b, paediatric, pharmacokinetic (PK) bridging study TEACH (*Vélez et al., 2022*) were not available before submission of the Watson et al. manuscript. However, in the TEACH study, in which the tafenoquine dosage in paediatric patients was chosen to match blood exposure in adults receiving 300 mg, tafenoquine was efficacious and generally well tolerated: no patients withdrew from the study because of adverse events (*Vélez et al., 2022*).
- The model used by Watson et al. to predict the recurrence-free rate at 4 months after a 450 mg dose is hypothetical and does not consider data regarding the tafenoquine exposure–response relationship. Importantly, tafenoquine exposure achieved with a single 300 mg dose approaches the plateau of the exposure–response curve; therefore, the incremental recurrence-free rate gained by the proposed 50% increase in dose is small and unlikely to be justified by overall benefit–risk considerations (*GSK, 2018b*). In addition, as primaquine and tafenoquine have different PK and metabolic profiles, we consider the extrapolation of data from primaquine to tafenoquine to be problematic (*Hounkpatin et al., 2019*; *Baird, 2019*).
- We feel that, overall, some of the conclusions do not acknowledge evidence-based safety concerns for a >300 mg dose of tafenoquine and do not consider additional data from the INSPECTOR study that the recurrence rate of *P. vivax* infection within 6 months of tafenoquine treatment was not significantly affected by bodyweight (*Sutanto et al., 2023*).

Watson et al. mentioned the phase 2b dose-selection study (DETECTIVE) of tafenoquine (*Llanos-Cuentas et al., 2014*), from which a single 300 mg dose co-administered with chloroquine was chosen for phase 3 evaluation in adults. However, they did not point out that, in this study, exposure was a significant predictor of efficacy and doubling the tafenoquine dose from 300 mg to 600 mg was associated with only a marginal increase (from 89.2% to 91.9%) in the primary efficacy endpoint, relapse-free efficacy at 6 months (*Llanos-Cuentas et al., 2014*). Moreover, in addressing the INSPECTOR study of tafenoquine in Indonesian soldiers, Watson et al. did not specify that this was a study of tafenoquine administered with an artemisinin-based combination therapy rather than chloroquine and, as such, is not directly comparable due to poorly understood but confirmed interactions impacting tafenoquine efficacy (*Sutanto et al., 2023*). Watson et al. also suggest that tafenoquine 300 mg is likely inferior to 'optimal primaquine regimens', but it is unclear whether such regimens are the WHO-recommended schedules of primaquine or regimens defined as optimal based on non-regulatory studies of primaquine. Watson et al. provided no specific reference or dosage characterising optimised primaquine therapy, so this *a priori* inferiority cannot be evaluated.

## Results
### Efficacy models employed by Watson et al
In their efficacy models, Watson et al. explored the association between the odds of *P. vivax* recurrence and the following predictors: mg/kg dose of tafenoquine; $AUC_{0-\infty}$; peak plasma tafenoquine concentration; terminal elimination half-life; and Day 7 methaemoglobin (MetHb) level. However, details of how the best predictor was selected and how statistical significance was judged were not provided in the manuscript.

### Use of a 4-month versus 6-month follow-up period
Regarding radical curative efficacy, Watson et al. selected *P. vivax* recurrence within 4 months as their primary endpoint. However, the trial-defined primary endpoint at 6 months from the pivotal tafenoquine clinical trials (*Llanos-Cuentas et al., 2019*; *Llanos-Cuentas et al., 2014*; *Lacerda et al., 2019*) was an FDA requirement and was mandated for analysis purposes. This was to maximise the probability of capturing relapses, including those from regions with longer latency periods. Watson et al. used the INSPECTOR study (*Sutanto et al., 2023*) as one of two reasons to justify the selection of a 4-month endpoint. Relapse rates differ greatly from country to country, so the duration of the endpoint should not be based on rates observed in a single country. Moreover, the 6-month rate of loss to follow-up (only 9.1%) does not justify a change of treatment endpoint from 6 months to 4 months . Furthermore, Watson et al. describe a possible association between tafenoquine mg/kg dose and the odds of recurrence (using logistic regression), with a 4-month rather than 6-month follow-up. An odds ratio of 0.66 (95% confidence interval [CI]: 0.51, 0.85) is cited by Watson et al. in their analysis of the effect of tafenoquine mg/kg dose in patients who received tafenoquine 300 mg, but descriptive details for this result and the analysis are limited. Figure 2 in the Watson et al. manuscript shows Kaplan–Meier survival curves for time to first recurrence, based on tafenoquine mg/kg dosing category, but some areas require clarification, such as how the dosing bands were selected.

### Rationale for tafenoquine dose selection
Importantly, the classification and regression tree analysis, in which a clinically relevant breakpoint tafenoquine AUC value of 56.4 μg·h/mL was identified, was not discussed (*Tenero et al., 2015*). Population PK modelling revealed that tafenoquine 300 mg would provide systemic exposure greater than or equal to the AUC breakpoint in approximately 93% of individuals, who would have a high probability (85%; 95% CI: 80, 90) of remaining relapse-free at 6 months (*Tenero et al., 2015*). Therefore, this '… model-based approach was critical in selecting an appropriate phase 3 dose' for tafenoquine (*Tenero et al., 2015*). Although data from the TEACH paediatric study (*Vélez et al., 2022*) were not available when Watson et al. conducted their analysis, had the data been available, they would have validated the AUC approach to tafenoquine dose selection, with an overall efficacy of approximately 95% (*Vélez et al., 2022*). Individuals (aged 2–15 years) were given tafenoquine, based on body-weight, to achieve the same median AUC as the 300 mg dose in adults (children weighing >10–20 kg received tafenoquine 100 or 150 mg; >20–35 kg received 200 mg; and >35 kg received 300 mg). The

recurrence-free rate at 4 months was 94.7% (95% CI: 84.6, 98.3) (*Vélez et al., 2022*), and the TEACH study supported the successful approval of tafenoquine for children aged 2–16 years by the Australian TGA in March 2022 (*Medicines for Malaria Venture, 2022*).

Another important counter to the mg/kg-based dose selection is that, when bodyweight categories were fitted as a continuous variable in the INSPECTOR study (using data for the time to recurrence for all participants), neither bodyweight nor bodyweight-by-treatment interactions were statistically significant ($P=0.831$ and $P=0.520$, respectively) (*Sutanto et al., 2023*).

## Use of an unvalidated biomarker

The hypothetical causal model proposed by Watson et al. for the clinical pharmacology of tafenoquine for the radical treatment of *P. vivax* malaria is problematic. Central to this model are MetHb production and active metabolites. However, MetHb is not a validated biomarker of tafenoquine efficacy, and currently there is no evidence, from non-clinical or clinical studies, of circulating active metabolites of tafenoquine; if such metabolites were fleetingly present, they would require extraordinary potency to exert any significant pharmacodynamic effect (GSK Investigator Brochure. Data on file).

In addition, although Watson et al. state that increases in blood MetHb concentrations after tafenoquine administration were highly correlated with mg/kg dose, no correlation coefficients, indicating strength of correlation, were discussed in the manuscript. It should be re-emphasised that MetHb is not a validated, surrogate biomarker of antimalarial treatment efficacy as a radical cure for *P. vivax* malaria and was used as a safety measure in the INSPECTOR study (*Sutanto et al., 2023*).

## Potential safety concerns

In the *Tolerability and safety* section, Watson et al. state that severe haemolytic events were rare; however, this is because all the studies were randomised and controlled, which excluded patients with <70% G6PD activity. In addition, no mention was made that, in one of the constituent studies (which examined the dose–response for haemoglobin decline in participants with 40–60% G6PD enzyme activity) (*Rueangweerayut et al., 2017*), dose escalation of tafenoquine from 300 mg to 600 mg was not attempted due to safety concerns about potential haemolysis in patients with G6PD deficiency. In tafenoquine-treated patients in the real-world setting, some instances of severe haemolysis might be expected, and it is already known from the previously highlighted phase 1 study that the haemolytic potential of tafenoquine increases with increasing dose (*Rueangweerayut et al., 2017*). Watson et al.'s *Tolerability and safety* section also mentions that one tafenoquine-treated patient had a >5 g/dL decrease in haemoglobin level, but the baseline haemoglobin level and tafenoquine dose are not mentioned. The section may have benefitted from a holistic discussion of safety parameters per tafenoquine dose group: for example, the occurrence of serious adverse events, gastrointestinal adverse events (beyond the selective discussion of vomiting within 1 hour post dose) and neuropsychiatric adverse events.

## Discussion

Watson et al. conclude that 'the currently recommended adult dose is insufficient … increasing the adult dose to 450 mg is predicted to reduce the risk of relapse'; however, we have raised several concerns relating to these conclusions. In particular, we feel that the safety concerns associated with a higher-than-approved tafenoquine dose co-administered with chloroquine have not been thoroughly considered: the safety analysis is limited, and the increased risk of haemolysis in patients with G6PD deficiency that a 450 mg tafenoquine dose (which is 50% greater than the approved 300 mg dose) would pose in vulnerable populations in limited-resource settings is not adequately discussed. In some malaria-endemic regions, 8% of the population may be G6PD deficient, although wide variability exists, and in sub-Saharan Africa and the Arabian peninsula the prevalence of G6PD deficiency may exceed 30% (*Howes et al., 2012*; *P. vivax information hub, 2012*). Therefore, in regions with fragile healthcare systems and limited availability of relevant testing for G6PD deficiency, potential exists for a significantly increased risk of haemolysis if tafenoquine is administered at an above-recommended dose (450 mg). Importantly, off-label use of a dose not robustly evaluated in clinical trials would pose a considerable risk to patient safety.

Regarding tafenoquine efficacy, the rationale for a dose increase to 450 mg has limitations. Watson et al. suggest that a 50% increase in the adult dose of tafenoquine (from 300 mg to 450 mg) would prevent one relapse of malaria for every 11 patients treated. However, this number-needed-to-treat estimate is not balanced by a number-needed-to-harm estimate for acute haemolytic anaemia. In addition, the phase 2b part of the DETECTIVE study (*Llanos-Cuentas et al., 2014*) showed that, in countries where the trial was carried out, single doses of tafenoquine 300 mg and 600 mg co-administered with chloroquine had similar relapse-free efficacy at 6 months (89.2% and 91.9%, respectively); therefore, the lack of additional benefit for tafenoquine 600 mg in DETECTIVE and the phase 1 study, which demonstrated dose-dependent haemolytic potential for tafenoquine, favour a 300 mg dose.

In summary, based on currently available data, dosing tafenoquine at the approved 300 mg dose, in combination with chloroquine, carefully balances efficacy and safety in the radical cure of *P. vivax* malaria; indeed, tafenoquine 300 mg demonstrated a favourable benefit–risk profile in a comprehensive clinical development programme that included at-risk populations in regions with fragile or resource-restricted healthcare systems. Real-world monitoring of efficacy in geographic locations known to have greater recurrence rates with primaquine (e.g. in the Asia Pacific region with Chesson-type strains) will show if tafenoquine is prone to the same phenomenon. If higher doses are required in these regions or when co-administering with artemisinin-based combination therapies (not currently approved for use), due possibly to drug interactions, well designed, adequately powered clinical studies assessing safety and efficacy will be required before considering recommendations for label expansion. The arguments raised by Watson et al. come with the concerns articulated here, and we assert that a tafenoquine dose increase from 300 mg to 450 mg when co-administered with chloroquine is not supported by available fact-based evidence for the radical cure of *P. vivax* malaria in adults aged ≥16 years.

## Acknowledgements

Funding for this article was provided by GSK. Medical writing support was provided by David Murdoch, a contract writer working on behalf of Apollo, and Alex Coulthard of Apollo, OPEN Health Communications, funded by GSK, in accordance with Good Publication Practice 3 (GPP) guidelines (https://www.ismpp.org/gpp-2022).

## Additional information

### Competing interests

Raman Sharma, Chao Chen, Lionel Tan, Katie Rolfe, Ioana-Gabriela Fiţa, Siôn Jones, Anup Pingle, Rachel A Gibson, Panayota Bird: Employee of GSK; shareholder in GSK. Navin Goyal, Hema Sharma: Former employee and shareholder in GSK.

### Funding

No external funding was received for this work.

### Author contributions

Raman Sharma, Chao Chen, Ioana-Gabriela Fiţa, Siôn Jones, Anup Pingle, Rachel A Gibson, Panayota Bird, Formal analysis, Writing – original draft, Writing – review and editing; Lionel Tan, Katie Rolfe, Navin Goyal, Hema Sharma, Conceptualization, Formal analysis, Writing – original draft, Writing – review and editing

### Author ORCIDs

Lionel Tan https://orcid.org/0000-0002-3822-497X
Katie Rolfe https://orcid.org/0000-0002-8635-4891
Siôn Jones https://orcid.org/0000-0003-1649-8133
Anup Pingle https://orcid.org/0000-0002-9844-7323
Rachel A Gibson https://orcid.org/0000-0002-1338-1290
Hema Sharma https://orcid.org/0000-0001-6396-2344

Decision letter and Author response
Decision letter https://doi.org/10.7554/eLife.89263.sa1
Author response https://doi.org/10.7554/eLife.89263.sa2

---

## Additional files

### Supplementary files
• MDAR checklist

### Data availability
Data sharing is not applicable to this article as no datasets were generated or analysed.

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
