## [Decision Letter]

*In the interests of transparency, eLife includes the editorial decision letter and accompanying author responses. A lightly edited version of the letter sent to the authors after peer review is shown, indicating the most substantive concerns; minor comments are not usually included.*

Thank you for submitting your work entitled "Comment on 'The clinical pharmacology of tafenoquine in the radical cure of Plasmodium vivax malaria: an individual patient data meta-analysis'" for further consideration by eLife. Your revised article has been evaluated by Dominique Soldati-Favre (as Senior Editor), Urszula Krzych (as Reviewing Editor), and a Reviewer (who has chosen to remain anonymous).

Please consider the comments and suggestions made by the Reviewer (please see below) and respond accordingly. We will also write to you separately about some editorial issues that need to be addressed.

*Reviewer #1:*

I have reviewed:

i) the comment from Sharma et al at GSK about the original eLife paper by Watson et al

ii) the response from Watson et al to the comment from Sharma et al.

I think both letters should be published substantially as they appear currently.

I do not agree with everything in Sharma et al regarding their defense of the original dosing regimen of 300mg Tafenoquine (plus chloroquine). In particular, Sharma et al shows an acute (hyper) sensitivity to safety issues which should be moot if the drug is truly given only to those who have been tested for G6PD deficiency. To base one's argument solely on those who are failing to meet this stated standard seems unusually strict especially when there are valid arguments for a higher dose for efficacy.

Arguments about efficacy are largely limited to SE Asia/Oceania and I find it strange that neither letter states that tropical Asian vivax will require more tafenoquine than other geographic areas since this is well known with its primaquine predecessor. Using the total drug data set which includes mostly Latin American studies to argue against increasing the drug regimen in SE Asia seems disingenuous. I really think the arguments should be focused on the rapidly and frequently relapsing malaria of SE Asia as that is the actual clinical problem. Sharma et al should not be diluting the issue with efficacy data mostly from Latin America as that is not where the problem is located.

Methaemoglobin (MetHb) may not be a validated endpoint but it is an important indicator of redox active metabolism. GSK's initial error with tafenoquine was to regard the drug as not metabolized on the basis of in vitro liver microsome studies. The INSPECTOR study [1] clearly indicates no efficacy against relapse when tafenoquine is combined not with chloroquine, but artemisinin. MetHb is the critical marker, likely of drug metabolism to a redox active intermediate. I think it is highly likely that INSPECTOR reflects lack of metabolism to the 5,6 orthoquinone and substantially reinforces Watson's argument that more studies are needed.

As such additional studies are evidently in process (a DSMB is being formed), it seems that both letters should reflect that more data is needed specifically focused on areas with frequently relapsing vivax malaria (SE Asia) in order to better define whether a higher dose regimen of tafenoquine is indicated or not.

[1] Sutanto I, et al. 2023 The Lancet Infectious Diseases. DOI: https://doi.org/10.1016/S1473-3099(23)00213-X

*Editorial issues to be addressed:*

Further to the email below, here are the editorial issues that need to be addressed in the revised version of your submission. […]

b) You will see from the response from Watson et al that they claim your submission contains a number of factual errors about their paper. Please revise your submission to address the comments made on the following lines in Watson et al.

i) lines 61-63

ii) lines 86-89

iii) lines 170-175

iv) lines 187-190

v) Please also consider if you need to revise your submission in response to lines 97-110

---

## [Author Response]

Reviewer #1:I have reviewed:i) the comment from Sharma et al at GSK about the original eLife paper by Watson et alii) the response from Watson et al to the comment from Sharma et al.

The authors would like to thank the reviewer for their comments and feedback.

I do not agree with everything in Sharma et al regarding their defense of the original dosing regimen of 300mg Tafenoquine (plus chloroquine). In particular, Sharma et al shows an acute (hyper) sensitivity to safety issues which should be moot if the drug is truly given only to those who have been tested for G6PD deficiency. To base one's argument solely on those who are failing to meet this stated standard seems unusually strict especially when there are valid arguments for a higher dose for efficacy.

Tafenoquine in combination with chloroquine is indicated for the radical cure (prevention of relapse) of *P. vivax* malaria. For adults, adolescents, and children weighing >35 kg, a single 300 mg dose is recommended, and is approved in 8 countries. The reviewer asserts that there are “valid arguments for a higher dose for efficacy”. However, a multicentre, double-blind, randomised, Phase 2b dose-selection study of tafenoquine plus chloroquine found no “evidence for any increase in efficacy between tafenoquine 300 mg and tafenoquine 600 mg” (Llanos-Cuentas et al, Lancet, 2014).Recently, the Tafenoquine Roll-out STudy (TRuST), a non-interventional, observational study evaluated the feasibility of implementing a new treatment algorithm for *P. vivax* radical cure with quantitative, point-of-care G6PD testing, and single-dose tafenoquine or primaquine within the Brazilian public health system. TRuST reported superior effectiveness (possible recurrence/relapse) for Tafenoquine vs. Primaquine at day 90 based on routine patient records from the Brazilian malaria epidemiological surveillance system (SIVEP-Malaria), collated retrospectively for all consenting patients ≥6 months old (with parasitologically confirmed *P. vivax*) (TRuST abstract presented at ASTMH 2023).

The reviewer also notes that the authors of the letter to the editor are being “hyper” sensitive to the potential risk of harms to prospective patients. The claim is based on the premise that all potential tafenoquine recipients will be tested for G6PD deficiency and that tafenoquine will then be withheld from all those deemed G6PD deficient. Of course, this would be correct if 100% of potential tafenoquine patients were tested for G6PD deficiency and that 100% of those deemed G6PD deficient were excluded from receiving tafenoquine. Unfortunately, this is not what is likely in the real world as MMV, in collaboration with the Brazilian Ministry of Health, demonstrated in TRuST; as mentioned above, this study assessed whether a new treatment algorithm for managing radical cure of *P. vivax* malaria according to a patient’s age and G6PD activity could be successfully implemented in Brazil’s Amazon region. All patients were assessed for G6PD deficiency, and none were treated with tafenoquine based on unknown G6PD activity. However, 16 (0.6%) patients were inappropriately treated with tafenoquine despite having inadequate G6PD activity (<70% of normal). The reviewer is reminded of the work of Rueangweerayut et al., 2017 who demonstrated that haemolysis in G6PD-heterozygous females was greater in those with lower G6PD enzyme activity levels. As other malaria-endemic countries approve and provide access to tafenoquine, the reviewer is also reminded that the pharmacovigilance infrastructure in such countries may be less well developed than in Brazil and that the prevalence of G6PD deficiency in such countries may also be much higher than in Brazil. For this reason, the authors feel justified in erring on the side of caution in their consideration of the benefit-risk of 300 mg tafenoquine with chloroquine in potentially vulnerable populations, given the evidence from multiple clinical trials and the recent real-world observations of efficacy and safety for this treatment regimen.

Arguments about efficacy are largely limited to SE Asia/Oceania and I find it strange that neither letter states that tropical Asian vivax will require more tafenoquine than other geographic areas since this is well known with its primaquine predecessor. Using the total drug data set which includes mostly Latin American studies to argue against increasing the drug regimen in SE Asia seems disingenuous. I really think the arguments should be focused on the rapidly and frequently relapsing malaria of SE Asia as that is the actual clinical problem. Sharma et al should not be diluting the issue with efficacy data mostly from Latin America as that is not where the problem is located.

Despite evidence that tropical Asian vivax malaria is linked to a higher probability of relapsing malaria than Latin American strains for primaquine historically, this does not affect the safety questions related to tafenoquine administration in populations living in Asia. As broadly described in different sections of the manuscript, the 300 mg dose of tafenoquine administered with chloroquine currently offers the most balanced option in terms of efficacy and safety where the drug indicated for the treatment of the blood stage of Pv malaria is CQ.Use of a dose 50% higher than the 300 mg dose approved by the US FDA would significantly increase the risk of adverse effects, as noted above, where TQ is given by mistake to G6PD deficient individuals in populations where the prevalence of G6PD deficiency may be higher than others and pharmacovigilance infrastructure may be less well developed. As described in Llanos-Cuentas et al, Lancet, 2014, 300 mg tafenoquine could be reaching the dose-efficacy plateau as efficacy is similar to that observed with the 600 mg dose. Although not available at the time of writing of the Watson et al. article, in the TEACH paediatric study which had study sites in Vietnam (overall 53% of participants were of Southeast Asian ethnicity) and median exposures matching the 300 mg adult tablet dose with chloroquine, the 4-month recurrence-free efficacy was 94.7% (Vélez ID, et al. Lancet Child Adolesc Health 2022;6(2):86–95; https://doi.org/10.1016/s2352-4642(21)00328-x).

Reports describing higher efficacy of primaquine when high-dose primaquine is used could be related to its pharmacokinetics (PK), different metabolic profile and its short half-life where time above a concentration threshold for efficacy maybe be prolonged (for PQ active metabolite) by increasing parent drug dose; this may not necessarily be true for tafenoquine which has a more favourable PK profile and longer half-life. Based on the current evidence tafenoquine 300 mg with chloroquine currently offers a balanced safety/efficacy relationship where the drug indicated for the treatment of the blood stage of Pv malaria is CQ.

Real-world evidence of lack of efficacy 300 mg tafenoquine with chloroquine in SE Asian populations after roll-out is required before concluding that the tafenoquine dose is inadequate in these regions. If an increase in real-world treatment failure rates were observed in certain regions, then well designed, adequately powered clinical studies assessing the safety and efficacy would have to be conducted before considering recommendations for label expansion for use of doses higher than 300 mg. The authors have added the following statement on monitoring of efficacy in geographic locations known to have greater recurrence rates with primaquine:

“Real-world monitoring of efficacy in geographic locations known to have greater recurrence rates with primaquine (e.g in the Asia Pacific region with Chesson-type strains) will show if tafenoquine is prone to the same phenomenon. If higher doses are required in these regions or when co-administering with ACTs (due possibly to drug interactions), well designed, adequately powered clinical studies assessing safety and efficacy will be required before considering recommendations for label expansion.”

Methaemoglobin (MetHb) may not be a validated endpoint but it is an important indicator of redox active metabolism. GSK's initial error with tafenoquine was to regard the drug as not metabolized on the basis of in vitro liver microsome studies. The INSPECTOR study [1] clearly indicates no efficacy against relapse when tafenoquine is combined not with chloroquine, but artemisinin. MetHb is the critical marker, likely of drug metabolism to a redox active intermediate. I think it is highly likely that INSPECTOR reflects lack of metabolism to the 5,6 orthoquinone and substantially reinforces Watson's argument that more studies are needed.

The authors feel it is important to note that GSK does not solely base its conclusions on metabolism observed from in vitro studies. We have also undertaken non-clinical and clinical in vivo studies. In particular, semiquantitative mass balance clinical studies showed drug-related material (DRM) identified in blood and plasma was almost exclusively in the form of unchanged tafenoquine. All other circulating components observed were minor, the most notable being a carboxylic acid metabolite, which represented ≤6% of the parent concentration. DRM was excreted very slowly in human urine and, consistent with animal studies, was primarily excreted as products of O-demethylation, oxidation, dearylation and glucuronide conjugation, all of which had been previously seen in both rats and dogs.With most of the drug remaining as parent drug, for a metabolite to have a pharmacological effect it would have to have extraordinary potency and be very fast-acting to be the main source of efficacy for tafenoquine.

Additionally, unlike primaquine, there is no clinical evidence that tafenoquine efficacy is negatively impacted by poor/intermediate CYP 2D6 metaboliser status, and tafenoquine potentially offers a therapeutic option to these patients. This difference in metabolic profile indicates that tafenoquine has a different route of metabolism to primaquine, therefore possibly a different relationship to formation of MetHb. MetHb was originally designed as safety biomarker not as an efficacy biomarker – no studies have proven that production of MetHb is associated with an active metabolite for tafenoquine or required for efficacy.

Lack of efficacy has been demonstrated when tafenoquine has been coadministered with dihydroartemisinin-piperaquine, an artemisinin-based combination therapy used as an alternative to chloroquine in areas of chloroquine resistance (Sutanto et al, Lancet Infect Dis, 2023). Results from the INSPECTOR study have been preliminarily recapitulated in non-clinical in vitro/in vivo studies with our collaborators which suggest an antagonistic pharmacodynamic interaction of tafenoquine + 4-aminoquinoline + dihydroartemisinin, not the lack of metabolism to the 5,6 orthoquinone (unpublished data). Furthermore, as noted in the INSPECTOR publication “An association between methaemoglobin and efficacy following treatment with primaquine has been previously noted; however, in post-hoc analysis combining data from this study and previous tafenoquine phase 3 studies, there was no significant association between maximum methaemoglobin and efficacy (p=0·257)”.

Any prospective use of higher doses of tafenoquine with ACTs to try to overcome this apparent antagonism would have to be supported by well designed, adequately powered clinical studies assessing safety and efficacy before considering recommendations by regulators for label expansion for use of doses higher than 300 mg with ACTs. The authors have updated the manuscript to specifically refer to tafenoquine “co-administered with chloroquine” where appropriate and state in the discussion section that well designed, adequately powered clinical studies assessing safety and efficacy are required before considering recommendations for label expansion for use doses higher than 300 mg with ACTs.

As such additional studies are evidently in process (a DSMB is being formed), it seems that both letters should reflect that more data is needed specifically focused on areas with frequently relapsing vivax malaria (SE Asia) in order to better define whether a higher dose regimen of tafenoquine is indicated or not.

The authors again thank the reviewer for their feedback. As per comment 3, the authors have added a statement to note that monitoring of efficacy in geographic locations know to have greater recurrence rates with primaquine is required.

Editorial issues to be addressed:

b) You will see from the response from Watson et al that they claim your submission contains a number of factual errors about their paper. Please revise your submission to address the comments made on the following lines in Watson et al.i) lines 61-63

We believe that this opinion is not supported by the facts. Importantly we did not state “that a higher 450 mg dose should be recommended”, but rather that “clinical trials of higher tafenoquine doses are needed to characterise their efficacy, safety and tolerability”.Tafenoquine in combination with chloroquine (CQ) is indicated for the radical cure (prevention of relapse) of *P. vivax* malaria. For adults, adolescents, and children weighing >35 kg, a single 300 mg dose is recommended, and is approved in 8 countries.

The reviewer asserts that their “results demonstrate clearly that when using the current 5 mg/kg regimen a substantial proportion of adults will be under-dosed, and therefore that there would be a substantial benefit from increasing the dose to 7.5mg/kg" equivalent to higher 450 mg dose; but data currently available do not support this assertion. Recently, the Tafenoquine Roll-out STudy (TRuST), a non-interventional, observational study evaluated the feasibility of implementing a new treatment algorithm for *P. vivax* radical cure with quantitative, point-of-care G6PD testing, and single-dose tafenoquine or primaquine within the Brazilian public health system. TRuST reported superior effectiveness (possible recurrence/relapse) for tafenoquine vs. primaquine at Day 90 based on routine patient records from the Brazilian malaria epidemiological surveillance system (SIVEPMalaria), collated retrospectively for all consenting patients ≥6 months old (with parasitologically confirmed *P. vivax*) (TRuST abstract presented at ASTMH 2023).

As global RCTs have demonstrated the effectiveness of a single 300mg dose of tafenoquine (+ standard doses of CQ) for radical cure of *P. vivax* malaria in multiple countries, with TRuST confirming the effectiveness of the treatment regimen in a real-world context, it is not currently possible to conclude that the approved regimen is inadequate. If real-world treatment failure rates were observed with the approved regimen, then we would agree with Watson et al. that well designed, adequately powered clinical studies assessing the safety and efficacy of higher doses of tafenoquine should be conducted.

Lack of efficacy has been demonstrated when tafenoquine has been co-administered with dihydroartemisinin-piperaquine, an artemisinin-based combination therapy used as an alternative to CQ in areas of CQ resistance (Sutanto et al, Lancet Infect Dis, 2023). Results from the INSPECTOR study have been preliminarily recapitulated in nonclinical in vitro/in vivo studies with our collaborators which suggest an antagonistic pharmacodynamic interaction of tafenoquine + 4-Aminoquinoline + dihydroartemisinin (unpublished data).

We have changed “that a higher 450 mg dose should be recommended” to “could reduce *P. vivax* recurrences substantially”. The authors have furthermore, added a statement in their discussion stating “Realworld monitoring of efficacy in geographic locations known to have greater recurrence rates with primaquine (e.g. in the Asia Pacific region with Chesson-type strains) will show if tafenoquine is prone to the same phenomenon. If higher doses are required in these regions or when co-administering with artemisinin-based combination therapies (not currently approved for use), due possibly to drug interactions, well designed, adequately powered clinical studies assessing safety and efficacy will be required before considering recommendations for label expansion”.

ii) lines 86-89

Lines 86–89: “Sharma et al. state that “Details of how the best predictor was selected and how statistical significance was judged were not provided*”.* The code for the statistical analysis is openly accessible. For the main analysis it is provided as an RMarkdown file (https://github.com/jwatowatson/Tafenoquineefficacy):

We agree that the code is available and has been sign-posted in Watson et al., however, pinpointing the exact code pertaining to selection of the best predictor isn’t obvious as the code is 1299 lines long. Additionally, given that univariate and multivariate analysis performed the choice of predictor and judgement of significance cannot be readily identified from the code.

Methodological details important to assessing how the analysis by Watson et al. was performed would perhaps be best discussed and presented upfront in the manuscript. We have amended our manuscript to state “Details of how the best predictor was selected and how statistical significance was judged were not provided in the manuscript”.

The author’s thank Watson et al. for highlighting the line 981 and line 1023, these appear to be univariate regression models that account of baseline parasitaemia and study site e.g.

mod_auc_all = stan_glmer(outcome_primary~ scale(AUC) + logpara0 + (1|studysite),family='binomial', data =outcome_dat,verbose=FALSE,refresh=0)

whereas in the manuscript under the “Comparison of Predictors of Recurrence” Watson et al. state “To determine the main predictors of recurrence, we fitted a multivariable penalised Bayesian logistic regression model. This included the following variables as predictors: the tafenoquine mg/kg dose; the plasma tafenoquine AUC[0,-); the plasma tafenoquine Cmax; the day 7 MetHb concentration (%);the terminal elimination half-life t1/2 ; and the baseline parasite density.” We believe more clarity on which method was used and model selection criteria would be very informative. Perhaps a sentence in the methods section could clarify this point.

iii) lines 170-175

Lines 170–175: “Sharma et al. state:“increases in blood MetHb concentrations after tafenoquine administration were highly correlated with mg/kg dose, no correlation coefficients were presented.” This is incorrect. We quantified the correlation between the tafenoquine mg/kg dose and the day 7 methaemoglobin using the linear model coefficient: “Each additional mg/kg was associated with a 19% (95% CI: 17% to 21%) increase in day 7 MetHb concentrations.” The strength of the correlation is also made clear from Figure 3c”.

There appears to be a miss-understanding of the terminology, Watson et al. do state the numerical relationship between mg/kg dose (i.e. 19% increase in d7 MetHb for every mg/Kg unit) based on their regression model.

This, however, is not the correlation coefficient. Commonly, when discussing strength of correlation one would use the Pearson correlation coefficient (r) or coefficient of determination (R^2^). When one examines the Rcode for this relationship the Multiple R-squared is reported to be 0.365 which may be considered a weak-to-moderate correlation regardless of statistical significance. The authors have changed the text to read “no correlation coefficients, indicating strength of correlation, were discussed in the manuscript.”(*iv) lines 187-190*

Lines 187–190: “Sharma et al. state that “off-label use of a dose not robustly evaluated in clinical trials would pose a considerable risk to patient safety”. We have not recommended “off-label” use of higher doses of tafenoquine. Our main recommendation was that clinical trials of higher doses should be conducted with the aim of providing optimal efficacy in patients at greatest risk of relapse”.

In an idealised situation therapeutic agents would never be used off-label. Unfortunately, in real-world clinical situations, given Watson et al.’s recommendations there is a realistic risk that some clinicians/practitioners will presume that 300 mg will not be efficacious and seek to administer a larger dose off-label. As asset owners of tafenoquine, it is important that we highlight the risk of off-label use.

As commented above, real-world evidence of lack of efficacy 300 mg tafenoquine with CQ in certain geographic locations after roll-out is required before concluding the tafenoquine dose is inadequate in specific regions. If an increase in real-world treatment failure rates were observed in certain regions, then well designed, adequately powered clinical studies assessing the safety and efficacy would have to be conducted before considering recommendations for label expansion for use doses higher than 300 mg.

As mentioned above, the authors have added a statement in their discussion stating “Real-world monitoring of efficacy in geographic locations known to have greater recurrence rates with primaquine (e.g. in the Asia Pacific region with Chesson-type strains) will show if tafenoquine is prone to the same phenomenon. If higher doses are required in these regions or when co-administering with artemisinin-based combination therapies (not currently approved for use), due possibly to drug interactions, well designed, adequately powered clinical studies assessing safety and efficacy will be required before considering recommendations for label expansion.”

v) Please also consider if you need to revise your submission in response to lines 97-110

Lines 97–110: “The optimal duration of follow-up in a study of radical curative efficacy remains debated. In most tropical regions *P. vivax* relapses are highly predictable and occur within a few weeks to months after initial treatment. Trials which have included no 8-aminoquinoline treatment arms indicate that >90% of first relapses occur within four months (Commons et al., 2023). Longer follow-up increases sensitivity (more relapses included) but lowers specificity (more reinfections included). The results of our analysis are almost identical when a 6-month endpoint was applied. This was shown in Figures 4&5, Appendix 1 in Watson et al. (2022). Sharma et al. consider that we provide insufficient detail of the sensitivity analysis which included patients receiving 300 mg only. The sensitivity analysis is given on line 501 in the section “Logistic Regression” in the main RMarkdown (TQ_efficacy.Rmd). The same primary analysis was performed on the restricted dataset of patients who received a 300 mg dose of tafenoquine. Sharma et al. ask how the dosing bands were chosen for our Figure 2. The selected bands were chosen for simplicity of visualisation ensuring enough patients fell into each category and were thus meaningful. Importantly no quantitative results depend upon this categorisation.”

We highlight in our letter that the FDA recommended the use of the 6-month follow-up period and this was to maximise the probability of capturing relapses, including those from regions with longer latency periods. We would therefore still recommend the original endpoint criteria in each trial be maintained.

The authors thank Watson et al. for specifying the line number in the code that shows the sensitivity analysis. This is helpful as it is not immediately apparent in the 1299 lines of code, as commented above these details would be very informative in the original manuscript in the appropriate section.

The authors thank Watson et. al for the information regarding the selection of dose bands in Figure 1, which may have been useful in the legend of Figure 1. Watson et. Al state no quantitative results depend on this categorisation, do the Kaplan-Meier survival curves change significantly with different categorisations?

For additional consideration:

Response to other comments in Response to comment on “The clinical pharmacology of tafenoquine in the radical cure of Plasmodium vivax malaria: an individual patient data meta-analysis" not solicited by editor:

Lines 47–49: “The dose-response relationship for tafenoquine-induced haemolysis in G6PD deficiency has not been well characterised (the data available comprise 3 heterozygous females with >40% enzyme activity who were given a 300mg single dose, Rueangweerayut et al., 2017)”:

The entirety of the evidence was considered in the above study, not just the 300 mg dose group. Groups of six female G6PDheterozygous subjects with an enzyme activity range 40–60% of the adjusted site defined median value for G6PD-normal males were administered 100 mg and 200 mg also. The total number of 15 females with G6PD-heterozygous status were administered tafenoquine. Furthermore, “compared with G6PD-normal subjects, the maximum decrease in hemoglobin in G6PD-heterozygous subjects was slightly greater for tafenoquine 100 mg, but this difference became more pronounced at tafenoquine 200 mg and was evidently greater with the 300 mg dose, with dose-limiting toxicity occurring in 3/3 subjects”. which was the stopping criteria of a dose group.

Lines 51–52: “Compared with primaquine, the most widely used 8-aminoquinoline antimalarial, tafenoquine is fundamentally a more dangerous drug.”:

Tafenoquine has the same haemolytic risk as standard dose primaquine, if used appropriately. It is true that if an acute hemolytic anemia is observed exposure to the tafenoquine cannot be stopped due to it's long half-life in comparison to primaquine but the statement by Watson et al. is too general and may be taken out of context. Could Watson et al. slightly amend the statement to:

“tafenoquine, if not used appropriately, may have a higher risk of causing adverse consequences"?

Lines 69–72: “The DETECTIVE phase 2 trial was small. Only 57 patients received 300 mg and 56 received 600 mg, distributed across four different countries. One of the countries was India where long latency relapse strains are found. These relapses would have occurred after 6 months (the study’s follow up duration).”:

We agree that the long latency relapse strain in India resulted in very few relapses being observed within the 6-month follow-up period in any of the dose groups. For this reason, the analysis of relapse-free efficacy over 6 months follow-up was repeated, excluding participants recruited at the Indian sites. As would be expected, the relapse-free efficacy estimates decreased in all treatment groups, with those in the tafenoquine 50 mg, 100 mg, 300 mg and 600 mg being 49%, 48%, 87% and 90% respectively, but the interpretation remained the same as that from the analysis including Indian participants. There was little difference between the 300 mg and 600 mg dose groups.

The study was designed to have sufficient power to test for the superiority of each dose of TQ+CQ versus CQ alone, tested in a sequential manner. As Watson et al. correctly point out, it wasn’t designed to have sufficient power to test for a difference between TQ dose groups. However, the observed difference in efficacy-free relapse rates between the 300 mg and 600 mg dose groups, both including and excluding Indian participants was small (<3%), and the 95% confidence intervals were largely overlapping. The difference was also much smaller than that observed between TQ+CQ and PQ for the 300mg TQ dose group (>11%), so the difference between the 300 mg and 600 mg dose groups is not considered to be clinically relevant.

Lines 74–76: “We fit a logistic regression model to the DETECTIVE phase 2 efficacy data. We estimate an odds ratio for any recurrence at 6 months of 0.67 per mg/kg increase (95%CI 0.58 to 0.76). At 4 months this odds ratio is 0.62 (95% CI 0.51 to 0.72), i.e. almost identical to our result from the pooled dataset (Watson et al., 2022).”:

Are these results available to view?

Lines 80–82: “A primaquine total dose of 7mg/kg is approved by WHO for the Southeast Asian region. A pooled individual patient data meta-analysis shows that this dose is clearly more efficacious than 3.5 mg/kg”:

Tafenoquine has a very different metabolism and pharmacokinetic profile in comparison to primaquine; therefore, findings from this study cannot be directly applied to tafenoquine. As stated previously, the authors have now stated in the manuscript if an increase in real-world treatment failure rates were observed in certain regions, then well designed, adequately powered clinical studies assessing the safety and efficacy would have to conducted before considering recommendations for label expansion for use doses higher than 300 mg.

Lines 114–122: “The relevance of the quoted 56.4 *μ*g.h/ml threshold for AUC is unclear. We do not see why this threshold as determined should be considered “clinically relevant”. A CART model determines optimal breakpoints by minimising a given loss function. It appears from the cited publication (Tenero et al., 2015) that the authors used the default parameters in the rpart function in R (from the rpart package). If this is correct, then it would imply that the CART model was fit using the Gini criterion, thus maximising the “purity” of the split (recurrence vs no recurrence). The Gini criterion is not an appropriate choice. Under the Gini criterion the cut-off threshold is the value which equally balances sensitivity and specificity for a continuous covariate which has a continuous relationship with the outcome (e.g. a linear relationship on the log-odds scale).”:

A simple plot of AUC/relapse demonstrates that at 300 mg dose we are effectively at the top of the dose response curve. The dose-ranging study may not have involved thousands of subjects but provide very relevant data to identify how increasing exposure impacted recurrence rates in a GCP study. Watson et al’s analysis uses the same data therefore will be subject to the same limitations in terms of dynamic range and granularity dosing and exposures available. We therefore need to have more real-world data to see if 300 mg is insufficient. Our 300 mg dose provides exposures of approximately 2X of the CART 56.4 ug.h/ml for more than 90% of subjects at the 300 mg dose (~100 ug.h/mL) which indicates exposures achieved with 300mg would probably be robust to a change exposure threshold.

Lines 139–144: “Sharma et al. quote the TEACH study (paediatric tafenoquine study) as validation of the AUC approach to tafenoquine dose selection (Vélez et al., 2022). Firstly, we note that TEACH was a single arm, non-randomised study with no control group. Hence it is unclear how “efficacy” is defined. Secondly, we note that the mg/kg doses in TEACH were higher on average than in the adult efficacy studies. Using the reported mean body weights (Table 1 in Vélez et al., 2022), the mean dose was approximately 7.0 mg/kg.”:

Vélez et al., 2022, clearly states recurrence-free efficacy was 94·7% as defined in https://clinicaltrials.gov/study/NCT02563496?tab=table

Indeed, according to the allometric relationship between clearance and body weight, paediatrics would require higher mg/kg dose than adults to achieve comparable AUC. Analysis of the TEACH data reveals that 100 mg, 200 mg and 300 mg dose groups had mg/kg dose ranges of 5.0-8.3, 5.7-9.5 and 4.8-8.5, respectively. It must be noted, as discussed above, when translating to pediatric doses applying allometric principles shown to hold in the population pharmacokinetic modelling of tafenoquine, giving paediatrics the same mg/kg generally results in significant underexposure (AUC).

Furthermore, three recurrences occurred in the study with weight-normalised doses of 5.7, 7.5 and 6.1 mg/kg in the 100 mg, 200 mg and 200 mg dose groups. respectively. These data, despite being limited, did not show that lower mg/kg doses were associated with higher recurrence.